# Mechanochemical Synthesis of Polyanilines and Their Nanocomposites: A Critical Review

**DOI:** 10.3390/polym15010133

**Published:** 2022-12-28

**Authors:** Cesar A. Barbero, Diego F. Acevedo

**Affiliations:** Research Institute for Energy Technologies and Advanced Materials (IITEMA), National University of Río Cuarto (UNRC)-National Council of Scientific and Technical Research (CONICET), Río Cuarto 5800, Argentina

**Keywords:** mechanochemistry, conductive polymers, polyanilines, nanocomposites

## Abstract

The mechanochemical synthesis of polyanilines (PANIs), made by oxidative polymerization of anilines, is reviewed. First, previous knowledge of the polymerization reaction in solution is discussed to understand the effect of different parameters: oxidant/monomer ratio, added acid, oxidant, temperature and water content on the properties of the conducting polymers (molecular weight, degradation, doping/oxidation level, conductivity, and nanostructure). The work on mechanochemical polymerization (MCP) of anilines is analyzed in view of previous data in solution, and published data are critically reconsidered to clarify the interpretation of experimental results. A key factor is the production of acids during polymerization, which is often overlooked. The production of gaseous HCl during MCP of aniline hydrochloride is experimentally observed. Since some experiments involves the addition of small amounts of water, the kinetics and heat balance of the reaction with concentrated solutions were simulated. A simple experiment shows fast (<2 min) heating of the reaction mixture to the boiling point of water and temperature increments are observed during MCP in a mortar. The form and sizes of PANI nanostructures made by MCP or solution are compared. The extensive work on the production of nanocomposites by MCP of anilines together with different nanomaterials (porous clays, graphene, carbon nanotubes, metal, and oxide nanoparticles) is also described.

## 1. Introduction

Polyanilines are conducting polymers [1], with a wide range of technological applications: electromagnetic shielding [2], electrochromics [3,4], photothermal antitumoral/antibacterial therapy [5], electromagnetic actuators [6], electrode materials of batteries [7,8], supercapacitors [9], flexible electronics [10], conducting hydrogels [11], electrochemically-driven ion exchangers [12], fluorescent nanoparticles [13], electrochemical sensors [14], etc. The synthesis of the polyanilines is usually made by oxidative polymerization of anilines in solution [15], at a relatively low concentration (0.1–0.2 M) [16]. Therefore, the synthesis of polymer uses a lot of water, which becomes contaminated with byproducts (e.g., benzidine which is a known carcinogen). This is a relevant matter in the chemical industry. For example, using the IUPAC standard procedure for polyaniline (PANI) synthesis [17], which uses a 0.2 M concentration of aniline chlorhydrate (AniHCl)), ca. 54 liters of water (assuming 100% yield) have to be used to produce 1 kg of PANI (Emeraldine Base, EB). As we will see below, the maximum concentration cannot be increased because the reaction is highly exothermic, and relatively low (>50 °C) temperatures cause degradation. One way to avoid solvents is the use of mechanochemical synthesis where two solids are brought to a reaction under mechanical action (milling, grinding). Engineering and environmental considerations favor MC synthesis over the same reaction in solution. If the yield and conversion are high, reactor size is much smaller in MC, allowing large scale synthesis in small factories. 

Mechanochemistry (MC) is a method with a long history [18], which has received renewed interest in recent years, especially in the field of polymer synthesis (MCP) [19]. Besides reacting solids, modern work in mechanochemical synthesis (MS) includes the use of small amounts of liquid (liquid assisted grinding, LAG [20]), solid diluents [21], or even additional polymers [22] to produce the reaction. Of great interest are those materials which cannot be made by a solution reaction but occur under MCP [23]. However, only one mechano-exclusive synthesis, that is, a conducting polymer only produced by MCP, has been reported [24]. On the other hand, oxidative polymerization of anilines is an exergonic reaction (exothermic) where no external energy input (e.g., mechanical) is required. Moreover, in the case of oxidative polymerization of aniline, to give polyaniline it is usually performed under cooling (0–5 °C) since at higher temperatures additional reactions occur which could degrade the conducting polymer. That is, energy is withdrawn from the reaction instead of introducing it. In MCP, the temperature is usually not controlled or measured [25], therefore, high temperature excursions could occur and the product could be degraded. The initial steps of aniline oxidation are well understood, but the details of polymer growth are unclear. The final product (conducting polymer) is insoluble in the solvent, and it is possible for the chains to grow in the solid state (a relevant point for solvent-free polymerization) or to grow in solution until it becomes insoluble and precipitate. Data on mechanochemical polymerization of anilines leading to conducting polymers are critically analyzed along with the effect of adding additives or changing the type or amount (relative to monomer) of reactants. The previously known mechanism in solution and the effect of parameters such as monomer/oxidant ratio, acidity, and the presence of precursor molecules (e.g., 4ADA) is used to analyze the mechanochemical data. A special case, widely studied in the field, involves the formation of composites with different solids. Here, the mechanical force could induce the milling of the solid (e.g., graphite to graphene), and the polymerization occurs in the exposed surface allowing producing nanocomposites. However, in some cases the polymer is already formed and the mechanical force allows the milling and mixing of both materials. In any case, the method is a simple way to produce nanocomposites containing PANIs. Finally, a future outlook in the field is discussed. IUPAC defines a mechanochemical (MC) reaction as a “chemical reaction that is induced by the direct absorption of mechanical energy” [26]. There are two ways to approach the MC technique. The physicochemical approach studies systems where mechanical energy (including static pressure) act at bond level to produce the reaction. On the other hand, the organic synthesis approach uses mechanical processes (grinding, milling, pressure, ultrasonic, etc.) as a tool to: (i) induce reactions not achievable by other methods (thermochemistry, photochemistry, and electrochemistry); (ii) improve the synthesis in terms of yield, reaction time, purity of the product, etc.; or (iii) make the synthesis more sustainable by eliminating (or diminishing drastically) the use of solvents. The synthetic approach, which is used in all the work reviewed here, is a practical one. The goal is the product, not the knowledge of the mechanism. In that sense, it is analogous to other modern tools of synthesis (microwave, ultrasound, radiofrequency, or hydrothermal) whose mechanism is still in discussion but are routinely used in organic chemistry. A better understanding of the parameters affecting the reaction should allow setting the best conditions to obtain desired outcomes: smaller time, highest yield, and absence of side reactions. Side reactions are more relevant in polymer than in general organic chemistry because the side products remain usually linked to the polymer. Experimental aids, such as the use of drops of liquid to aid the milling (LAG), have a profound effect on the synthetic output. The liquid could lubricate the grinding and transmit the pressure while being chemically inert. On the other hand, the liquid could dissolve one of the reactants, producing a heterogeneous (solution/solid) reaction. If all reactants are dissolved, the reaction is in solution but at a high local concentration. In both cases, the reaction rate obviously increases compared to a diluted solution. In the case of water, it not need to be added but could be present by absorption in hygroscopic reactants unless the reaction is performed in a dry box. The grinding process produces small (<1 μm) particles which have large surface areas. Therefore, water adsorption from the environment could be important. It is noteworthy that such considerations of the likely role of adventitious water were already taken into account early on (1893) by Johnston and Adams [27], while discussing the effects of even and uneven pressure on inorganic reactions. Another reactant, which could exist in normal MCP, is oxygen from air. We will see that authors propose reactions (albeit catalyzed and taking several days) where oxygen from air is the only oxidant. 

A novel way to do LAG involves using fused salts, ionic liquids (IL), or deep eutectic solvents (DES) as liquids [19]. As will be described in the review, some authors use fused salts, generally occurring by the heat of reaction, as an aid in MCP. Since the polymerization product is a solid polymer instead of a macromolecule dissolved in a solvent, the aggregation form (i.e., the nanostructure) produced in the synthesis remains in the final solid. It is noteworthy that the most often used method to produce PANI nanofibers involves oxidative polymerization at the interface of a solution of aniline in a solvent (immiscible with water, e.g., CHCl_3_) and an aqueous (acid) solution of the oxidant (APS) [28]. The nanofibers grow towards the aqueous solution. The heterogeneous reaction between solid anilinium salt and solid oxidant resembles closely the liquid/liquid interphase. As will be seen, some authors indeed obtained PANI nanofibers by MCP. 

The use of mechanical actions, being in a mortar/pestle, ball mill, tumble mill, or extruder, is known to influence the yield of product in synthetic chemistry. Moreover, experimental parameters, such as the number and size of balls, the hardness of them, and the effect of the ball/jar material (e.g., steel), could also affect the reactions. However, no comprehensive model exists to account for such effects, which have to be ascertained for each reaction. As will be seen below, the vast majority of polyanilines syntheses use mortar and pestle as MC. It is known that grinding in a mortar has low reproducibility, is limited to shorter times (although 6 h grinding times have been reported), and could endanger the operator through the release of toxic fumes/gases. On the other hand, if a visual confirmation of reaction exists (e.g., color change), it allows using only the necessary time. This is likely the reason that using mortar and pestle is the technique of choice in the synthesis of inorganic complexes, organic charge transfer complexes, and synthesis of conducting polymers, where clear color change occurs. In closed systems (e.g., ball mill jars), the reaction could be carried out to for long times to assure complete conversion. However, polymer chain scission by mechanical action is well known [19], and ball milling of already formed polymers could decrease the molecular weight. On the other hand, closed milling devices allows for maintaining an inert (oxygen- and water-free) atmosphere, by filing the closed jar in a glove box. The closed container is also relevant when possible toxic and/or corrosive fumes are produced during reaction. There are several theories to explain mechanochemical effects in different areas of chemistry: (i) the hot spot theory, (ii) the magma-plasma model, and (iii) the pseudo-fluid model [19]. None of them is able to explain mechanochemical polymerization reactions completely [19]. Moreover, other processes occur during MC experiments that do not relate to mechanical energy input but influence the reaction. Milling/grinding achieves an intimate mixing of reactants at the nanometric scale (likely involving single or few macromolecules), allowing them to react. The process is dynamic, as size diminution creates fresh reactive surfaces, which allow reaction between solids and/or create small domains of highly concentrated solution in small amount of solvents (e.g., water). The later can be adsorbed from the reaction atmosphere or is added in LAG. Moreover, particle diminution decreases diffusion limitations, allowing movement of ions inside the solid during reaction time (min to h). Mechanical stress during milling/grinding generates heat at the interface between particles, and local temperature increases can lead to large increases of the reaction rate. While most reactions are endergonic and use the mechanical energy as input, the IUPAC definition of MC includes exothermic reactions between two solids where the reaction should occur spontaneously (if the reaction rate is large enough) but is inhibited by the absence of contact (i.e., occurs at the interface, which is small for macroscopic solids). Upon particle size decrease and/or pressuring in MC processes, the reaction occur. However, the reaction should stop when grinding is stopped. In fact, the well-known and thermodynamically favorable “termite” reaction of aluminum powder with Fe_2_O_3_ is initiated usually by external heating. The same reaction became self-sustainable and controlled by grinding under MC conditions [29]. As we will see, the polymerization of ANIHCl with APS at high concentrations resembles the “termite” reaction (obviously at lower temperatures), and the models used to understand one reaction could be applied to the other. 

## 2. Results

### 2.1. Synthesis of Polyanilines

#### 2.1.1. Mechanochemical Polymerization of Aniline 

First, we discuss the state of the art in the polymerization of aniline in solution. Aniline is easily polymerized in an acid solution to produce PANI [30]. The formation of dyes and precipitates (likely polymers) was already produced by the oxidation of aniline even before Staudinger’s macromolecular hypothesis was defined. Therefore, they were considered compounds with only eight benzenoid rings. Wilstater and Dirogi [31], and Green and Woodhead [32], oxidized aniline (in acid media) using dichromate, persulfate, H_2_O_2_ (catalyzed by Fe^+3^), and ClO_3_^−^ (catalyzed by copper or vanadium salts). When metal containing ions were used, the product was heated with sulfuric acid to eliminate the metal ions [31], but cross-linking and/or degradation could occur. Josefowicz and coworkers [33] already recognized polyaniline as a semiconducting polymer, studied it, and built applications, all at least 9 years before the official discovery of conducting polymers [34]. They use ammonium persulfate to avoid metal contamination (or chlorination with chlorate), and it has become the most widely used oxidant [21]. The mechanism of aniline polymerization is supposed to involve three stages: an induction period, the growth of the polymer, and termination [35]. However, the stages occur simultaneously for all (or most) of the chains, and the termination occurs when the oxidant is consumed. This is unlike chain polymerization of vinyl compounds (radical, cationic, or anionic), where initiation, propagation, and termination occur along the whole polymerization time. Electrochemical (oxidative) polymerization involves similar steps, but the conditions are somewhat different [36]. The oxidant is the electrode, the oxidation occurs at the polymer/solution interface, and the mass transport conditions imply low concentration of the monomer. Indeed, when polyaniline is polymerized at high current densities, the polymer becomes degraded (showing the so-called “middle peak”) since oxidized polymer reacts with water. A similar situation exists when aniline is polymerized chemically (with oxidant) at ratios of oxidant/monomer larger than the stoichiometric amounts. 

The initial stages of aniline oxidation are as follows:

The aniline cation radical is produced by oxidation of unprotonated aniline, which is produced by fast equilibria from the protonated aniline (anilinium ion) (Figure 1). The initial condition is acidic (due to hydrolysis of the anilinium ion of added acid). Since the formation of the radical cation (oxidation) is fast and irreversible, all the anilinium could be converted to radical cation. The aniline radical cation suffer fast (<1 ms) follow-up irreversible reactions [36] to give 4-aminodiphenylamine (head-to–tail dimer) and benzidine (tail-to-tail dimer) (Figure 2). A proton is produced in the dimerization process, furtherly decreasing the pH. A solution of 0.2 M anilinium ion (IUPAC, [21]) will have a pH of ca. 2.64, which will decrease with the reaction progress. In nonaqueous (and non-protic) media (e.g., DMSO), all aniline is available for oxidation, but the aniline will deprotonate the growing chain, converting it into emeraldine base (EB). Chemical polymerization could proceed, but electrochemical polymerization will stop since EB is non-conductive and no electrons can flow through. However, in protophobic media (e.g., ACN), the protons produced are able to protonate the PANI film, maintaining it in its ES (conductive) form [37]. 

There is spectroscopic (in situ FTIR) and electrochemical evidence of the occurrence of those reactions [38], in the same conditions (acid aqueous media) used for the electrochemical and chemical polymerization of aniline. Benzidine can be produced by tail-to-tail dimerization of cation radicals or by the reaction of an aniline radical (produced by deprotonation of the cation radical). Since the cation radical is less acidic than the anilinium ion, it will remain protonated in acid media, becoming deprotonated in neutral media. Therefore, benzidine concentration increases at a lower pH [35]. The polymerization is carried out in acid media to avoid head-to-head coupling, which gives rise to hydrazobenzene and azobenzene (in basic media) [39]. Unlike 4ADA and benzidine, PANI could not grow from azobenzene molecules, and the polymer does not grow in basic media. However, if the media is un-buffered (such as in solvent free polymerization) the production of protons could lower the pH (at least locally) and allow polymer grow. Additionally, nucleophilic addition of non-protonated (at higher pH values) aniline [40] gives rise to phenazine-like units (Figure 3) [41]. Indeed, this reaction (using 4ADA as precursor and dichromate as oxidant) is the method to synthetize molecular dyes with phenazine units [42], related to the first organic synthetic dye, mauveine [43].

The phenazine units localize the charges and lower its mobility, decreasing drastically the conductivity and acting as defects. The side chain could act as crosslinking points increasing the molecular weight of the polymer. Due to the existence of Peierls’s transition [44,45], the charges could not move along the whole PANI chains. Therefore, long chains (with a polymerization degree larger than 30) do not show higher intrachain conductivity. Moreover, longer chains will be difficult to assemble in order, and the interchain conductivity would be smaller. Therefore, promoting the crosslinking does not produce a more conductive polymer. The long discussion in the early 1900s [31,32,33] about greenable aniline black (likely linear emeraldine) and ungreenable aniline black (likely a polymer containing a large amount of phenazine units) suggests that phenazine units are produced by the reaction of aniline with emeraldine, using dichromate as an oxidant. Since dichromate ion is a weaker oxidant (Eo = 1.33 V_NHE_) than persulfate (Eo = 2.01 V_NHE_), the reaction does not seem related to polymer overoxidation. It is noteworthy that electrochemical polymerization is usually performed at a lower potential of 1.0–1.05 V_NHE_ [46]. It is known that pernigraniline (produced at such electrode potentials) reacts easily with nucleophiles [47], including water (with degradation). It is surprising that the chemical oxidant most widely used to produce PANI has a redox potential ca. 1.0 V higher than the one used in electrochemical polymerization. Chloride is the most widely used counterion to produce PANI. On the other hand, the formation of Cl_2_ by the oxidation of Cl^−^ with persulfate is thermodynamically favorable. Indeed, the usual polymerization procedure calls for APS to be dissolved in a small amount of water (maximum concentration = 3.5 M) and not in HCl solution (e.g., 1 M) because a strong chlorine smell is observed in the latter case. The Cl_2_ formed could produce the electrophilic substitution of aniline and/or of PANI. Moreover, solution polymerization is carried out at low temperatures (0–5 °C) under cooling, since at higher temperatures the nucleophilic attack of water on the polymer leads to the hydrolysis of the quinonimine units, yielding carbonyl groups (quinone like) [48]. All reactants are in solution, but the actual growth of the polymer chain is a heterogeneous reaction because the polymer is insoluble in water. That is, the oxidant could oxidize the polymer chain not at the growing end but in other part of the polymer (not even of the same chain), which is electrically connected to PANI. On the other hand, Stejskal and coworkers [49] have demonstrated that aniline could polymerize even at low temperatures (−27 °C) with solvent (water) in the frozen state. This is a solid state polymerization, similar to mechanochemical synthesis, while the heat release could produce local melted regions. The polymerization reaction is highly exothermic, and it is often done in a reactor cooled by ice. Cavallo et al., simulate the polymerization kinetics of aniline in aqueous acid [50], using the kinetic model described by Tzou and Gregory [51]. Using the reaction enthalpy and the thermal transfer parameter of the reactor (extracted from the cooling profile after the reaction has finished), it is possible to extract the kinetic constants from the temperature–time profiles. Moreover, the model can be used to simulate the temperatures and reaction times. At high effective concentrations (>1 M), the model will predict high temperatures and fast reactions. We simulate the kinetics of aniline (as AniHCl) polymerization with persulfate (equimolar to aniline) at the maximum concentration possible (3 M, due to solubility). The model suggests temperatures as large as 600 °C and initiation times of less than 10 s. To test the hypothesis, we experimentally mix 2 M solutions of AniHCl and APS (ca. 15 mL each), while measuring the temperature. Since the reaction heat could project corrosive solution driven by the water vapor, small volumes (<20 mL) should be used, and the observation and temperature measuring should be performed away from the direction of possible projections. After 10 s, the clear solutions (T_initial_ = 22.3 °C) turn blue, and the temperature increases, reaching 30 °C (after the induction period) at 40 s. Then, the temperature reaches ca. 100 °C after 115 s (ca. 2 min). After that, the water evaporates, and the temperature remains constant but likely does not reflect the temperature of the whole polymer mass. (Figure 1). Vapor is seen coming out, and a bluish-violet porous mass of polymer remains in the beaker. After cooling, water is added, and the polymer filtered out. These results show that the reaction media of concentrated solutions produces large amount of heat, as predicted by the model, being able to reach high temperatures (ca. 580 °C) in ca. 180 s (3 min) (Figure 1). The experimental data could not surpass the boiling point of water since it evaporates (absorbing heat), and the reaction was shut off for lack of solvent. The data are relevant to the analysis of mechanochemical synthesis where water is present since it isabsorbed from the atmosphere or added as part of the experimental procedure. Obviously, in real conditions the reactants and products will decompose below 580 °C. However, the data suggest that high (even 100 °C) local temperatures can be attained. 

The large surface area of the powder allows for heat release by radiation, air convection, and transference to the reaction holder. In an open mortar, the most common device used (see Table 1), even local water evaporation could reduce the temperature. In some cases, the reaction holder (e.g., mortar) was cooled before grinding (see Table 1). Given the large mass ratio between the grinding media and the sample, this procedure should maintain a low temperature. 

A simple test consisting the mixing of equimolar AniHCl and APS (both dried in a desiccator after separate milling) is devised (see SI for experimental details). It is observed that 5 min of milling is enough to change the temperature, while the color of the powder changes from pale yellow to dark blue (Figure 2). 

The temperature (measured by an infrared thermometer) changes from 27.3 °C to 31.2 °C. When only APS is grinded alone in the same mortar, no increasing trend is observed, suggesting that mechanical heat is not the cause of the increase of temperature. If the mortar is left still in a camera with a drying agent (anhydrous CaCl_2_), the temperature only decays due to cooling (to 28.9 °C). On the other hand, if the mortar is then placed in a chamber with 100% humidity, the temperature increased again to 29.8 °C, suggesting that the absorbed water from the environment promotes further reaction. 

While cryogenic grinders have the capability of cooling the grinding jar to low temperatures [52], ball mills were usually used at ambient temperatures to polymerize aniline. It is noteworthy that, in aqueous solution polymerization, relatively large temperatures (>50 °C) are usually precluded to avoid the nucleophilic attack of water [53]. On the other hand, it has been found that dry PANI backbone can withstand up to 500 °C without degradation [54]. Therefore, water-free MC polymerization of aniline should be better than that using liquid-assisted MC (Table 1). The stoichiometry of the reaction of oxidant (e.g., APS) and aniline (as anilinium salt) is shown in Figure 4 [55]: 

The reason that more moles of oxidant (25% of APS (2 e^−^/mol)) are required to produce PANI is that 0.5 moles of e^−^ are required (per PANI unit of four rings) to oxidize PANI to its pernigraniline state (from the ES units). During each monomer addition cycle, the product is formed in the emeraldine state, which is less reactive towards aniline. Potentiometric measurements show that the chain grows in the pernigraniline state [56]. If the oxidant is present at a molar amount of less than or equal to 2.5 times the moles of aniline, the reaction stops when the oxidant (limiting reagent) is used up and the chains remain in the emeraldine state. On the other hand, if excess (>2.5 more moles of APS than aniline) oxidant is present, the polymerization stop when all aniline (flimiting reagent) is used, but the chain stays in the pernigraniline state. In an aqueous solution, it suffers nucleophilic addition of water (>55 M), forming -OH terminated chains (aminophenol units) and/or cutting the chain forming the same groups. The excess oxidant reoxidizes the aminophenol units to quinonimine, and the degradation continues until all excess oxidant is used up. In electrochemical driven oxidation, applying a potential where emeraldine is oxidized to pernigraniline (>1.0 V_NHE_) induces the complete degradation of PANI into benzoquinone (detected by in situ FTIR in the solution [38]). One additional question regards the protonation state of the growing chain (i.e., present in the protonated state (pernigraniline salt, PNS) or an unprotonated state (pernigraniline, PN)). The electrochemical data of ion exchange suggest that PANI (PNS) deprotonates in 1 M monoprotic strong acid [57]. Since it is known that nucleophilic addition is faster in the protonated quinonimines than in the deprotonated ones [58], the nucleophilic attack of aniline to the growing chain (PN) will be faster at a higher acid concentration. This seems to be the case for [HCl] > 1 M [59]. 

The induction time occurs because dimers (4ADA and benzidine) are slowly formed from aniline to initiate the chains. The addition of the dimers or PANI eliminate the induction period [51]. Moreover, if other polymers containing aniline moieties are present in the polymer solution, the PANI chains grow linked to the polymer instead of becoming free chains [60].

The polymerization reaction produces protons, which form acid with the anions present [61]. Indeed, the release of a highly concentrated solution of acid has been observed during the solid state polymerization of aniline [62]. Usually, it is assumed that the anion in the anilinium salt is used as a counterion to dope the polymer. However, it is likely that both sulfate (from persulfate) and the anion (X^−^) in anilinium salt are used. In the solution, this could be the situation if XH is as strong an acid as H_2_SO_4_. However, if anilinium salts (e.g., CH_3_COO^−^AniH^+^) of weak acids are used, the anions are protonated, and only sulfate will be the counterion of PANI (ES form). Moreover, it is known that a high concentration of chloride ion (2–6 M) favors the nucleophilic addition of Cl^−^, leading to the formation of a polymer including 2-chloroaniline rings [63]. In a solution, such situation occurs at concentrations of HCl higher than 1 M. In a solid state reaction, pure HCl will be released, but in a wet reaction mixture the acid concentration could reach maximum solubility (ca. 12 M), then react with PANI.

While mechanochemistry can be traced back to prehistory, mechanochemical synthesis of PANI (called aniline blacks) also has a long history. Production of aniline black was a well-established process in the printing of cotton fabrics [64]. The process, discovered by J. Lightfoot, involves mixing AniHCl with chlorate salts and a catalyst (usually copper salts) [64]. The process was simple and provides a highly sought black color for inexpensive cotton cloth. The color was permanent and resistant to alkali (used in the 19th century to wash clothes) and light. The aniline blacks were specifically for cellulosic materials (cotton, paper, and wood) since they do not dye wool or silk well. It is likely that reducing groups (e.g., thiols) in the protein backbone of wool or silk inhibit the reaction. In our (UNRC) teaching labs, where wood is still used as counter material, Lightfoot’s mixture (as a paste) was impregnated in the wood and left to react for 48 h, providing a black dye resistant to chemicals. Since a paste of solid salts is used, Lightfoot’s recipe for dying cellulosic materials probably was the first industrial mechanochemical synthesis of (we know now) a conducting polymer. 

Table 1 summarizes the research on the polymerization of anilines (without other components). 

**Table 1 polymers-15-00133-t001:** Summary of reaction conditions for MCP of aniline.

Monomer	Oxidant	Acid Added	Mechanochemical Device/Material	Assistant Liquid	Time Pol./Curing	Washing	Ref
ANI	APS	H_4_SiW_12_O_4_	Mortar/unknown	none	30 min	H_2_O	[65]
AniH_2_Sul AniCSA	APS	H_2_SO_4_ CSA	PBM 600 rpm/SS jar and balls	none	1 h	EtOH/H_2_O	[66]
Ani (added acid)	APS	TsOH DBSH MSH	Mortar	H_2_O	30 min	DEE/EtOH /H_2_O	[67]
Ani (added acid)	APS	HCl H_2_SO_4_ H_3_PO_4_	Mortar	H_2_O	20 min	DEE/EtOH /H_2_O	[68]
AniHCl ^1^	APS	HCl	Mortar/agate	none	1.5–4 min/24 h	H_2_O	[69]
AniH_2_Sul	APS	H_2_SO_4_	Pan mill(steel) Mortar	none	20 cycles 40 min	H_2_O/EtOH	[70]
AniHCl	APS	H_2_SO_4_	Mixing mortar + rest	none	5 min mix/24 h	EtOH	[62]
AniHCl	FeCl_3_	HCl	Mixing mortar + rest	none	5 min mix/1 week	EtOH	[62]
AniHCl	AgNO_3_	HCl NO_3_H	Mixing mortar + rest	none	5 min mix/1 week	EtOH	[62]
AniHSu	SeO_2_	SeO_3_H_2_	Mortar	none	1 h/ 24–48 h	H_2_O/Acetone	[71]
AniH_2_Sul	KPS	CSA CA	Mortar	none	30 min/ 24 h	H_2_O/MeOH	[72]
AniHCl	APS	HCl	PBM 300 rpm/agate	none	1 h	EtOH/H_2_O (pH = 2)	[73]
Ani_2_SA	APS	CA	Mortar	none	30 min/24 h	H_2_O/EtOH/MeOH	[74]
AniHCl	FeCl_3_·6H_2_O	HCl	Mortar	none	50 min	H_2_O	[75]
AniHCl	FeCl_3_·6H_2_O	HCl	Mortar	none	42 min	H_2_O/EtOH	[76]
Anit-PASA AniPAMPS	APS	t-PASA PAMPS	PBM 300 rpm/agate jar and balls	none	1 h	Dialysis	[77]

^1^ The photographs in [69] show a light blue-green color of AniHCl, while it should be uncolored. Our own experience suggests that aged AniHCl spontaneously polymerizes in air, contaminating the salt with PANI.

The first modern use of mechanochemical methods to synthesis of polyaniline was performed by Gong et al., who produce PANI in the presence of silicotunsgtic acid [65]. They react frozen aniline (−20 °C) with silicotungstic acid by MC, and then grind the product with APS in a mortar, producing PANI by MCP. They changed the ratio between the Keggin acid and APS, but they found and almost constant amount of anion (tungsten 50% in weight) and conductivity (ca. 0.3 S/cm), except in extreme values of the ratio. 

The foundational study of mechanochemical polymerization of aniline was published by Kaner and coworkers [66]. They polymerize aniline sulfate with ammonium persulfate (APS) using a sealed ball mill (stainless steel balls and jar). After 3 h of milling, the anilinium salt converts into polyaniline and the ammonium persulfate into ammonium sulfate. As in any mechanochemical reaction, a stoichiometric balance is important because there is no solvent to dissolve the other products. Use of aniline salts is reasonable since they are solid, but the pH of the media will be only moderately acid since the pKa of anilinium ion is 4.59 [78]. 

Therefore, even at a high concentration of 4.25 (solubility of the salt), the pH will be ca. 2. On the other hand, PANI in solution is usually polymerized in 1 M acid, with the pH close to zero. Kaner and coworkers studied the effect of Aniline/APS ratio on the yield of PANI (0.25 to 10). At a low ratio the yield is low (ca. 42%), and the maximum yield is obtained at a ratio 0.5 (65%) [66]. The yield decreases afterwards. The authors suggested that it is due to the formation of oligomers. However, it would imply that more than 80% of aniline was lost to secondary reactions, which is not detected experimentally during washing. The yield % should be calculated taking into account the limiting reagent (APS below ratio < 0.8 and anilinium for ratio > 0.8). Additionally, it has to be assumed that the mass of the polymer corresponds to the ES form of PANI (C_24_H_24_N)(SO_4_H)_2_). Otherwise, the yield will exceed 100%. The plot (Figure 3) gives a smooth variation where the highest yield % (for the limiting reagent) is 78.5% at a high anilinium/APS ratio (10). 

Since no solvent is present and the PANI chain grow by adding aniline units to the chain, it seems that a large excess of aniline is required to grow the polymer around small APS grains [66]. Moreover, the authors find out that the molecular weight increases when the ratio is changed from 1:1 (11,220) to 2:1 (18,200) and 3:1 (45,700) in a similar fashion to the recalculated yield (Figure 3) but not related with the published yield (Figure 4 in [66]), where a growing amount of oligomers is produced. The larger molecular weight reported (45,700 g/mol) is higher than the one reported before for polyaniline made in solution (27,000 g/mol), but the high MW peak (501,200 g/mol) is lower than the one measured before for PANI produced in solution (210,000 g/mol) [79]. An interesting point is the relatively large specific surface area (BET) of 69.7 m^2^/g obtained. It could be envisaged that oxidant (and/or ammonia sulfate product) crystals act as a pore former inside the polymer matrix. Another interesting result is the polymerization of aniline sulfate by simple hydrostatic pressure [66]. Monomer (aniline sulfate) and oxidant (ammonium persulfate) were finely milled and mixed by shaking. Then a high pressure (up to 16 kpsi) is applied between two steel rods. Since the product (PANI ES form) is electrically conductive and the reactants are not, the evolution of resistance (decreasing) shows the formation of the conducting polymer. The polymerization takes ca. 85 h but is only driven by pressure, not additional milling or local heating. The hydrostatic pressure system seems to be a very clean way to produce conducting polymers, but it has not been used after this work [66]. 

The reactants have to diffuse from the oxidant particle to the monomer particle and vice versa. The diffusion coefficients of ions inside crystals are in the order of 5 × 10^−13^ cm^2^/s [80]. The time required to polymerize by hydrostatic pressure would be satisfied by interdiffusion between particles of ca. 7 μm, which is a reasonable size of crystalline obtained by ball milling [81]. Moreover, in several published works on MC of PANI, a “rest” time period (1 day to 1 week) (Table 1) is applied after the grinding procedure. One likely situation is that water acts as a liquid mediator in the polymerization (Figure 2). Anilinium salts and APS are highly hygroscopic, and hygroscopic acids (H_2_SO_4_, HCl) are produced during the reaction. Therefore, water will be absorbed before, during, or after (the “rest” period) the reaction. In a particle of 10 μm, the ions should take 500 s (8.33 min) to interdiffuse (Fick’s law). This is much lower than the 1–3 h used in the ball milling [66], or the 83 h in the hydrostatic pressure experiment [66]. However, in another work (Table 1), the polymerization is performed in a mortar in less than 10 min. We try simple mortar grinding of dry AniHCl (1 g) and APS (1:1). The white powder mixture changes to green-blue after only 3 min of grinding, suggesting that the reaction occurs. Adding 50 mL of water (at the green-blue state) visibly induces a reaction showing a strong blue color of a solid dispersion. After 30 min a green solid appears, which, filtered, washed, and dried, shows the FTIR spectrum of PANI. However, after removing the solution with PANI, the flask (see Appendix A) shows a clear PANI film. It has been shown [82] that such film grows on the surface of the glass and cannot be produced by adsorption of solid PANI microparticles. Therefore, not all AniHCl and APS have reacted before water addition and the dilution converts the polymerization from a MCP to a solution/dispersion case. The point is relevant because when washing with water, a grinded mixture could induce a reaction. 

Therefore, the solvent used in washing is quite important (See Table 1). Otherwise, the mixture should be diluted greatly or quenched by freezing. In this case, no reaction is observed after water addition, and no film is formed in the beaker where water was added. Therefore, it could be assumed that the solid state reaction is complete. In a solid state reaction, if a non-volatile acid is used (e.g., H_2_SO_4_), the acid will remain with PANI and can be washed out with water. On the other hand, other acids such as HCl are volatile and will be released during the reaction. As can be seen in Appendix A, when AniHCl and APS are grinded in a glass mortar, the indicator paper shows a low pH (<1) since the wet indicator absorbs the HCl (gas) released by the polymerization reaction (Figure 4). 

The results suggest that the reactions in Figure 4 should be taken into account when analyzing data on MCP of aniline. Huang et al. use AniH_2_Sul as monomer salt and APS as oxidant [66]. Therefore, the only acid present is H_2_SO_4_. In MCP studies of aniline polymerization, other acids are added (Table 1), and even it is claimed that adding a weak acid (e.g., citric acid) “dopes” the polymer. In the presence of sulfuric acid, citrate is not present and could not dope PANI. Abdiryim et al. studied the effect of different strong organic acids (TSA, MSA, and DBSA) on the properties of solid state synthesis of PANI [67]. The procedure is different than that of Kaner and coworkers [66]. One ml of water is placed on a mortar, then pure acid (0.015 mol, 15 M) is dissolved and aniline (0.01 mol, 10 M) is added. As can be seen, there is a ca. 50% excess of acid. A white precipitate is formed, which is the anilinium salt of the acid since the concentration is larger than the solubility of the salt. Then, persulfate (2.2 g, 0.0097 moles) is added and ground in the mortar for only 30 min until the reaction is carried out. The ratio of aniline/oxidant is close to 1:1. As discussed before, the reaction does not occur in a dry state but in a concentrated (unknown) solution of anilinium salt and persulfate (maximum solubility 3.5 M). Therefore, it will be very fast (t_R_ = 60–100 s) and with a large thermal heating. Indeed, the cyclic voltammograms of PANI produced using this method show a significant “middle peak”, which is usually assigned to redox groups (e.g., quinones [61]), produced during PANI degradation. The same group studied the effect of different inorganic strong acids on the mechanochemical polymerization of aniline [68]. The procedure is similar, using 1 mL of acid solution (37% HCl, 96% H_2_SO_4_, and 87% H_3_PO_4_). The solid is filtered out and washed with diethyl ether (DEE), ethanol (EtOH), and water (W) and dried under vacuum. It should be mention that both APS, ammonium sulphate, and unreacted anilinium salts are insoluble in DEE; therefore, only oligomers could be removed. On the other hand, ethanol and water remove the organic salts, but usually ethanol is the last used. As can be seen, the most conductive polymer is PANI/HCl, while the least conductive is PANI/DBSA. Those with the highest yield have the largest conductivity. 

Bekri-Abbes and Srasra obtain a similar yield % (Figure 4) of AniHCl but lower (6000×) conductivity [69]. The effect of the aniline/APS ratio on yield % is quite different than the one shown in Figure 3. It is likely that the different method of MCP (closed ball mill in [66] and open mortar/pestle in [69] affect the MCP trough the absorption of water from air in an open mortar. 

Zhou et al. polymerize aniline sulphate with APS in a pan mill (600 rpm) and compared the polymer with one produced by 40 min of mortar grinding [70]. The molecular weight (estimated by viscosimetry) is lower for two pan mill cycles than for the mortar grinded mixture, becomes similar for 10 cycles, and is larger (>2×) for 20 cycles. Unfortunately, the time for the cycle is not provided. Sedenkova et al. polymerize AniHCl using different oxidants (APS, FeCl_3_, and AgNO_3_) [62]. The mechanochemical work is constrained to mixing the anilinium salt and the oxidant with a mortar (5 min). Then the powder is left to react in air (unknown humidity). Pellets are produced at 700 Mpa to measure conductivity, and a droplet of liquid is released. While APS produces PANI in 24 h, one week is required for Fe^+3^ and Ag^+^. While PANI is produced with APS, as expected, only oligomers and branched non-conductive polymers were obtained using Fe^+3^ or Ag^+^. The absence of polymerization was attributed to the low reduction potential of Fe^+3^ (0.77 V) or Ag^+^ (0.799 V). However, PANI has been produced in solution using iron chloride [83,84] and silver nitrate [85]. Additionally, Fe^+3^ [86] and Ag^+^ [87] can be used as oxidants to produce PANI in nanocomposites (see below). It is likely that longer (>5 min) grinding times are required to affect the polymerization. Posudievsky et al. oxidize AniHCl with APS in a PBM device (agate jar and mortar). They obtained a high conductive PANI (22.3 S/cm) and patented the procedure (see patents below) [73]. The procedure is quite similar to that used by Huang et al., except for using 300 rpm vs. 600 rpm in the ball mill and aniline hydrochloride instead of aniline sulfate [66]. The highly conductive PANI is obtained at a low APS/ANI ratio (0.5). The conductivity decreases when the ratio is increased and, at the stoichiometric ratio of 1.25, a polymer of relatively low conductivity (0.365 S/cm) is obtained. However, the conductivity for all materials is higher than the value reported by Huang et al. of ca. 0.01 S/cm [66]. Bhandari and Khastgir polymerize aniline in the presence and absence of citric acid (CA) to induce the formation of nanofibers of PANI [74]. While CA could have some effect on the morphology of PANI aggregates through hydrogen bonding, the authors claimed that PANI produced in the absence of CA is “undoped” when there are plenty of anions and acid to dope PANI. Moreover, the cyclic voltammogram (in 1 M H_2_SO_4_) for PANI produced in the presence or absence of CA show a broad peak with little separation between the first and second processes. This is not the typical behavior of “standard” PANI (synthesized in solution) and resembles PANI made electrochemically in nonaqueous solvent [88]. Du and coworkers polymerize aniline with Cl_3_Fe·6H_2_O (1:2 moles, 20% defect of oxidant) and found that the hydrated salt melts upon grinding and act as a liquid assistant or solvent [75,76]. Extensive hand grinding (40–50 min) produces a low yield (8%) of PANI in the form of nanofibers, such as those produced by interfacial polymerization [28,89], but not requiring toxic organic solvents (e.g., chloroform). Interestingly, the authors do not find any contamination of the PANI NF with Fe. Gribkova et al. use MC (PBM) to polymerize aniline as salts of polymeric acids, PAMPS and t-PASA [77]. The results were compared with the same salts used as monomers in solution. The results are similar and relatively low conductivities were obtained (Table 2). However, TEM images of the polymer composites show that the MC procedure renders layered arrangements of PANI–polyacid while the polymerization in solution gives globular structures. Recently Peramal et al. used SeO_2_ as an oxidant to polymerize aniline hydrogen sulfate [71]. They ground in a mortar for 1 hr and let the material rest for 24–48 h, observing the effect on polymer properties. They also changed the ratio of SeO_2_/ANI, but the authors do not declare which is the product of SeO_2_ reduction. SeO_2_ (in acid media) has a reduction potential of 0.74 V_NHE_, slightly lower than Fe^+3^ (0.77 V), which is able to polymerize aniline.
SeO_2_ + 4 H^+^ + 4 e^−^ = Se + 4 H_2_O(1)

However, the reaction (Equation (1)) produces Se, which is insoluble in water and probably remains inside the polymer. Since metallic Se has been used to “dope” PANI [90], increasing the conductivity (by three orders of magnitude), the fate of the Se seems relevant. However, the higher conductivity obtained is 0.488 S/cm (ratio 1:1.5 when waiting 24 h), which is lower than other PANI obtained by MC (Table 2). Therefore, no clear effect of Se on conductivity is found. SeO_2_ exchanges twice as many electrons/mol than APS. Therefore, if a 1:1 SeO_2_/ANI ratio is used, some of the SeO_2_ remains as SeO_3_H_2_ and is able to act as counterion dopant of PANI. However, since the first pKa of SeO_3_H_2_ is 2.62 [91], and the pKa of HSO_4_^−^ (present from the monomer salt) is 1.99 [91], the acid will be neutral (protonated), and most likely hydrogen sulfate will be the counterion dopant. Using an SeO_2_ ratio larger than 1 implies an excess of oxidant which could overoxidize the PANI. 

From the experimental point of view, 80% of the published MCP of aniline were made using a mortar and pestle. As was discussed in the introduction, the mortar and pestle is a low cost device to produce PANI in small quantities. Since the pale mixture at the beginning becomes dark blue upon polymerization, it is possible to grind the mixture until full conversion is achieved, making it reproducible. Using a clear glass mortar, it can be seen that the dark color begins to appear in the contact region between the mortar surface and the head of the pestle. This result suggests that the reaction is induced by mechanical grinding and not by the effect of oxygen/water, which will act more in the upper layer of the powder. On the other hand, a warning seems in order. A strong odor can be smelled during mortar grinding of AniHCl + APS, which could be due to volatiles (e.g., aniline) or aerosols of powdered reactants. Placing a filter paper, wet with a solution of p-benzenediazonium salt, close to the mouth of the mortar produces a deep yellow color likely due to the formation of the diazoaminobenzene from the reaction of the diazonium ion with aniline. Since aniline is toxic and a carcinogen, protective breathing devices and face masks should be worn when MCP of aniline salts is performed by open by mortar grinding. Another point involves the reactor material. Usually mortars are made of agate, which is quite inert. However, several authors do not state the material from which the mortar is made. While glass mortars seem as inert as agate, porcelain enamel erodes under grinding, baring the underlying porous ceramic. The pores of the ceramic retain by-products, polymer, and even cleaning agents, affecting following reactions. Stainless steel could corrode in concentrated acid, and soluble salts (e.g., Fe^+2^) catalyze the MCP reaction. 

As was discussed before, the aggregate form of PANI chains produced during polymerization create different shapes in the nanometer scale (nanostructures), which are preserved during washing/drying since the polymer remains insoluble. Figure 5 shows the SEM micrographs of PANI produced by solution polymerization and MCP (see experimental details in the Appendix A). 

While solution polymerization produces a fiber-like intertwined structure, MCP-produced polymer shows small nanoparticles (ca 70 nm). A statistical analysis of the SEM image (dark center region) of MCP PANI (Appendix A) allows one to obtain a distribution of nanoparticle sizes (Appendix A). The mean size is 70 nm. A geometrical calculation of the surface area gives 62 m^2^/g, which is close to the value (69.7 m^2^/g) measured by Huang et al. for MC-produced PANI [66]. If the particles could be deaggregated and stabilized, MCP could become a handy method to produce PANI nanoparticles. 

#### 2.1.2. Mechanochemical Polymerization of Substituted Anilines (with PANI-Like Linear Chains)

Anilines substituted in the *ortho* or *meta* position of the amino group could give head-to-tail and tail-to-tail coupling to produce dimers (similar to 4ADA and benzidine). From them, polyaniline-like chains can be grown Table 3 [61]. When the group is an electron donor, the reaction is faster than in the case of anilines. On the other hand, electron withdrawing groups slow down the chain growth, and only small oligomers (three to five rings) are formed. Therefore, copolymerization of aniline and the substituted (with electron withdrawing groups) aniline allows polymerization but with less than 100% of the rings substituted [92]. The incorporation of substituent groups in the ring has three effects: (i) to lower the conductivity, (ii) to increase solubility, and (iii) to increase the electrode potential for the oxidations: LE to ES and ES to PN. PANI is soluble only in concentrated acids (formic, acetic, and sulfuric) and some amides (NMP and DMF) when deprotonated or protonated with inorganic anions (e.g., Cl^−^). Introduction of anionic groups (e.g., –SO_3_^−^) makes the polymer soluble in basic media [93], while linking organic groups makes it soluble in organic solvents (e.g., toluene) [94]. It seems that the attached groups sterically hinder the π–π interaction between electron rings in neighboring chains, making the solid less stable and more soluble. The drawback is that such interactions are the basis for electron hopping between chains (orbital overlapping), giving bulk conductivity. Therefore, increased solubility means lower conductivity. Additionally, the presence of bulky substituents in one side of the aniline rings introduces steric distortions of the diphenylamine unit, decreasing the conjugation and the conductivity. However, when two identical groups are attached, the symmetry of the steric effect restores the planarity and the conjugation, with a restoration of conductivity [95]. The change in the electrode potentials of the LE-ES and ES-PN transitions is related to steric effects in the diphenylamine unit of PANI [96]. Small substituent groups (e.g., methyl) in the nitrogen (secondary amines) also allow dimerization, but benzidine-like dimer (tail-to-tail) formation is favored due to steric constraints of the head-to-tail attack [97].

Khadieva et al., polymerize aniline and poly[(N-2-hydroxyethyl)aniline by MCP [100]. They found that substituted aniline has lower conductivity. Abdiryim et al, polymerize o-toluidine by MCP using different organic sulfonic acids [101]. They found strong effect of the nature of the acid on conductivity

Large substituents in the nitrogen inhibit the formation of polymers. Table 3 summarizes the reaction conditions for the MC polymerization of substituted anilines. Jamal et al. use different organic strong acids in the polymerization of o-methoxyaniline (o-anisidine) [98]. They found evidence of branched polymer chains, as observed before in electrochemically polymerized oAS [102]. The conductivity depends of the dopant anion, varying between 0.001 and 0.01 S/cm, which is lower than PANI produced in the same conditions. Palaniappan et al. use MCP to produce poly(2,6-dimethylaniline) and obtain a material with conductivity comparable with PANI [99] (Table 4), while the polymer of aniline with only one substituent (P(oAS)) shows smaller conductivity. The polymers bearing electron withdrawing groups show even smaller conductivity (Table 4). As mentioned in the introduction, Modarresi-Alam et al. produce a homopolymer of 3-aminobenzensulfonic acid [24]. This is the only case where a polymer that is not possible to produce in solution (likely due to the effect of the electron withdrawing group) was produced by MCP. Moreover, they use a fused salt (Cl_3_Fe.6H_2_O at T > 45 °C) both as oxidant and solvent. The method in general would lead to the synthesis of a large variety of polyanilines. 

#### 2.1.3. Other Conducting Polymers Obtained from Substituted Anilines 

The MCP of substituted anilines discussed before occur in the reaction described in Figure 1, Figure 2, Figure 3 and Figure 4. However, some substituted anilines produce conducting polymers by other mechanisms. When aniline is substituted with –OH (aminophenols) or –NH_2_ (phenylenediamines) in the *ortho* or *meta* positions, the dimerization and grow steps occur in a way similar to PANI, but the chains could react intramolecularly to give ladder polymers with phenoxazine (aminophenols) or phenazine moieties in the chain [103]. Two possible polymers can be formed: one with linear chains (e.g., oAS) and the other with ladder chains made of phenoxazine (-O-) or phenazine rings (-N-) (Figure 5). The linear polymer should have a response similar to that of PANI, with two separated electron transfer processes. The electrochemical behavior of poly(aminophenol) and poly(o-phenylenediamine) suggests that the polymer is made of rings (ladder polymer), although it is possible that blocks of each type exists in the material. Moreover, it is known that the cyclic dimer (e.g., 2,3-diaminophenazine, DAP) is the main product of the chemical oxidation (with APS or Cl_3_Fe) of the substituted aniline (e.g., o-phenylenediamine) [104], and the electrochemical response of adsorbed DAP show the same electrode potential as the polymer. Therefore, even if a linear polymer is formed, it will form the ladder polymer by intramolecular cycle formation.

It has been suggested in [103] that a tail-to-tail (similar to benzidine) could be produced. In a similar fashion to aniline polymerization, the tail-to-tail dimer could link to two additional molecules of the monomer and initiate the polymerization. However, the electrochemical response of o-aminophenol or o-phenylenediamine show no peak system that could be assigned to tail-to-tail dimers [103]. Such peak systems exist and have been identified by IR spectroscopy in aniline oxidation [38]. While these ladder polymers show lower conductivity (2–5 orders of magnitude) than PANI [105], they are quite stable to degradation due to the ring chain structures [106]. Instead of the two separate electron transfers (LE->ES and ES-PN) of PANI, a single peak comprising two electron transfer is observed [107]. Diphenylamine (and triphenylamine [108]) could not produce head-to-tail chains but could polymerize by successive tail-to-tail reactions (such as dimerization to form benzidine) (Figure 6) [109]. 

In fact, diphenylamine is a well-known reactant for the detection of oxidants (e.g., dichromate), which, in diluted solutions, give tail-to-tail dimers with extended conjugations and visible colors [110]. The work on susbstitued anilines which show other polymerization mechanisms (Figure 5 and Figure 6) are summarized in Table 5. 

Zoromba et al. use MCP to produce poly(o-aminophenol). They compare MCP with interfacial polymerization [113]. Since the chemical polymerization of oAP in solution with APS has been described [114], it should be most suited to the task. The ratio of oxidant to monomer (1:0.81) suggests an excess oxidant. However, the mechanism (Figure 3) shows that double the amount of oxidant is required to produce ladder polymers. In the dimerization step, two moles of e^−^ (one mole of APS) are required per ring. But another 2 e^−^ are required for formation of the cycle, and 0.5 e^−^ to produce the polaron in the final polymer. On the other hand, both methods (MCP and IP) render a quite similar material. The same group pioneered the synthesis of a conducting copolymer: poly(o-aminophenol-co-(m-phenylenediamine)) [113]. Here, the feed ratio (oxidant/monomer) is 1:1.65. Therefore, there is a large excess of monomers. Using the same comparison criteria, the copolymers produced by MCP and IP are similar. In both case, the solid state method does not require handling of a toxic solvent (chloroform) used in IP. Palaniappan et al. produce poly(diphenylamine) (pDPA) by MCP, using Cl_3_Fe as the oxidant and diphenylamine as a monomer, using different acids (and none) added to the mixture [112]. The yield does not depend on the presence or kind of acid and is quite high (>70%). This is reasonable since the mechanism involves the dimerization of the radicals without a role for the nitrogen. The electrochemical response depends strongly on the acid used, suggesting different conductivity of the films. 

#### 2.1.4. Mechanochemical Synthesis of Nanocomposites

Mechanical mixing/grinding has been widely used to produce nanocomposites containing conducting polymers, being a dispersed component or a polymer matrix (Table 6). Since this review is focused on the MC synthesis of conducting polymers, we will discuss only the work where the conducting polymer is polymerized by MC, before or during the mixing with the other component of the nanocomposite. The fabrication of nanomaterials by MC have been reviewed before [115,116]. 

-Nanocomposites with clays

Bekri-Abbes and Srasra use MC (with APS) to polymerize clay (montmorillonite), where anilinium ions have been intercalated by cation exchange (using MC grinding) [118]. During the 20 min of MC grinding (clay + AniHCl), Mg^+2^, Al^+3^, and Fe^+3^ are exchanged by the anilinium ion. The ratio of AniHCl to the cation exchange capacity (CEC) of the clay was varied. The intercalation is verified by XRD. They found the oxidant/aniline ratio that gives the largest yield (86%). The material where the AniHCl amount used was equal to three times the CEC with an oxidant/aniline ratio of 0.75 gives the best results. It shows a low frequency conductivity of 5.7 × 10^−3^ S/cm, which is higher than an effective medium approximation of conducive PANI (σ = 3 S/cm) and dielectric MT would have predicted. It seems that aniline polymerizes in an ordered way inside the clay, being the lamellar structure of MT a template for the polymer. Bekri-Abbes and Srasra load montmorillonite with Cu^+2^ and grind for up to 20 min with AniHCl (depending on the ratio of Cu^+2/^MT to aniline) [123]. The highest DC conductivity of the NCs is 1.37 × 10^−5^. The same authors load MT with Fe^+3^ ions, and then grind the Fe^+3^/clay with AniHCl (120 min). However, to finish the polymerization, the mixture has to be left in the air for 10 days. The nanocomposite shows a DC conductivity of 0.058 S/cm, much higher than Cu^+2^/clay. The comparison of both publications supports the idea that the redox potential of the oxidant is not determinant in the chemical oxidation. While Cu^+2^/Cu^0^ redox potential is +0.34 VNHE, the Fe^+3^/Fe^+2^ redox potential is +0.77 V_NHE_. If the standard potential for oxidation of pernigraniline (the redox form during growing of PANI) is 1.05 V_NHE_, the polymerization should not occur (ΔE < 0 implies ΔG > 0). However, it is known that the emeraldine salt form is less reactive than pernigraniline towards aniline, but it could react [140]. Therefore, the chain could slowly grow at low electrode potentials. In fact, it has been shown that PANI can be synthesized by interfacial polymerization using air as an oxidant (Eo O_2_/H_2_O = 1.23 V_NHE_), albeit with catalysis by Cu^+2^ [141,142]. In a following work, Bekri-Abbes and Srasra use non-intercalated MT, which contains 6.3% Fe, as a catalyst for air polymerization of aniline [123]. AniHCl and virgin MT were grinded for 120 min and left to stand in the air for 14 days. The DC conductivity is 0.0016 S/cm. The relationship between conductivity and PANI content measured by Shakoor et al. [130] implies that the PANI content is below 15%, which is similar to other NC produced using Fe^+3^ as oxidant. Zheng et al. produce a NC of PANI and clay (kaolinite, KAO) using MCP [128]. Aniline was loaded inside the lamellar clay and then polymerized with APS by grinding. The basal spacing of the clay (measured by XRD) increases ca. 100% due to the intercalation of PANI chains. The thermal stability of intercalated PANI is higher than free polymer. The NC is applied as an anti-corrosion agent, dispersed in epoxy resin coatings. Yoshimoto et al. intercalate excess anilinium cations (from anilinium chloride) inside MT by MC. The basal spacing of MT increases ca. 157% [125]. Then, they use MCP (with APS) to produce a PANI(Cl-)/MT nanocomposite. The basal spacing of MT decreases to 38%, indicating the polymerization of aniline produces flat PANI chains. The thermal stability of PANI improves upon intercalation. The same group studied the formation by MCP of a PANI(SO_4_^−2^)/MT [126] and a PANI(p-TSO-)/MT [127] nanocomposite. In all cases, the thermal stability improves, and, in the case of PANI(p-TSO-)/MT [127], electronic conductivities of up to 0.01 S/cm are obtained. Kalaivasan et al. applied the same procedure to the synthesis of nanocomposites of MT with Ani, oTOL, and oAS [129]. The EPR studies indicate radical cations are present in the three NCs. Shakoor et al. [130] use a similar method to fabricate a PANI(TSO-)/MT nanocomposite. They found that conductivity of the NC increases proportionally to the amount of PANI in the NC where 99% PANI has a conductivity of ca 80% of a pure polymer. This value is reasonable since the electrons should percolate trough the nanocomposite, where only PANI is conductive. 

-Nanocomposites with carbon materials

Garcia-Gallegos et al. use MCP to produce nanocomposites of PANI and carbon nanotubes (CNTs, plain carbon (MWNT), and doped carbon (CNx)) [112]. They vary the amount of water added to the solid and the amount of CNTs. Very low yields (<3.15%) were obtained, and even lower yields (0.3%) were reported without liquid aid (water). It is noteworthy that Du et al., using FeCl_3_ to MCP aniline, also obtain low yields of PANI NFs (<8%) [75]. The electrical conductivity of the nanocomposites is similar to those NCs prepared by other methods, while the conductivity of pure PANI (using the same method) is quite low (ca. 10^−4^ S/cm) compared with other PANI prepared by MCP (see Table 2). The authors claim that they use an stoichiometric excess of oxidant (2:1), but the reaction stoichiometry (Figure 2) requires a 2.5:1 (Cl_3_Fe/AniHCl) molar ratio for complete conversion (equivalent to 1.25 in the case of APS). Jiang et al., uses mechanical grinding (2 h) to exfoliate graphite (using aniline as stripping agent) into graphene and then MCP (with APS) to produce PANI/G composites [133]. The TEM images show small nanoparticles of PANI (<20 nm) adsorbed on graphene plates. The content of PANI is up to 11.2%. The NC shows the characteristic cyclic voltammetry response of PANI, with a negligible “middle peak” revealing little degradation/crosslinking. The material shows a specific capacitance is up to 886 F/g. The capacitance has contributions of graphene (double layer capacitance) and PANI (pseudocapacitance). The method seems promising to fabricate supercapacitors, where graphene contributes with conductivity and surface area (double layer) and PANI with pseudocapacitance. PANI NPs (<20 nm) will have a surface area of more than 200 m^2^/g, contributing also to the double layer capacitance. Moreover, such small particles will switch in μs, even assuming a low diffusion coefficient (10^−10^ cm^2^/s) of ions inside PANI. Therefore, the material will have a large specific capacitance and fast response time. The MC technique is able to disperse the CNTs, and the presence of AniHCl protects the CNTs from mechanical damage due to the grinding, which occurs when the mixture (without monomer) is milled. Interestingly, the authors detect steel particles in the composite, coming from the milling jar. Jamal et al. synthesize PANI/TiO_2_/G and PANI/G by MCP of aniline [121]. The properties of the different composites are very similar. Du et al. use MCP to synthesize a NC of PANI and CNFs [122]. The CNFs become decorated with branched nanofibers as those reported before by MCP of Ani with FeCl_3_ [75]. The presence of PANI nanofibers in the surface of the CNFs aid the dispersion of the NC in ethanol. From the amount of CNF in the final solid (50%) and the mass ratio of aniline/CNF (1:0.2), it is possible to calculate a yield of ca. 8.3%, similar to the one reported before with the same oxidant [75]. It seems that MCP of PANI with FeCl_3_ as oxidant gives low yields [62,75,121,122]. Moreover, Modarresi-Alam et al. also obtain low yields of a substituted aniline (3ASA) in the same conditions [28]. Only using the melted (>37–45 °C) salt (FeCl_3_.6H_2_O) as oxidant allows them to obtain higher yields (46%). On the other hand, Koosheh et al. produce NC of silica with PANI using MCP (with FeCl_3_) [132]. They obtain large yields (up to 91%), confirming that in situ formation of NC is not equivalent to separate polymerization and mixing. The nature of the other component affects the polymerization mechanism. On the other hand, the conductivity of the materials is quite low (<0.0023 S/cm). The authors claim the reason is that FeCl_3_ (Lewis acid) is the sole dopant of the polymer. Such a hypothesis disregards the fact that protons are produced during polymerization (Figure 2) and chloride ions are produced during FeCl_3_ reduction. The HCl will protonate the EB form of the polymer creating protonic acid doping. Moreover, the ratio of oxidant/aniline is 2.5, which is just enough to polymerize aniline and oxidize the polymer to emeraldine. Therefore, no FeCl_3_ will be left to act as the Lewis acid dopant, and the reduction product (FeCl_2_) is a weaker Lewis acid. Abdiryim and coworkers produce nanocomposites containing PANI and MWNT [136,137], or SWNT [137]. The morphological studies (TEM) revealed that PANI/MWNT show rod-like and granular-like aggregates, while PANI/SWNTs did not display any features. A composite containing 16 wt% MWNTs show a specific capacitance (SC) of 522 F/g (measured by galvanostatic charge/discharge) [136]. When the nanocomposite (NC) with highest SC (16% MWNT, SC = 515 F/g) was compared with the best NC made using SWNT (8% SWNT, SC = 423 F/g), it is found that the NC with SWNT has a SC ca. 18% lower than the one with MWNT [137]. On the other hand, PANI/SWNTs exhibited higher cycling stability in neutral and alkaline electrolytes [137]. 

-Nanocomposites with photoactive materials

Yarmohamadi-Vasel et al. synthesize PANI nanofibers by oxidation with Cl_3_Fe in the presence of TiO_2_ NPs [117]. The nanocomposite show a conductivity of 0.79 S/cm which is higher than pure PANI NFs (0.09 S/cm) and TiO_2_ (1.5 × 10^−6^ S/cm). It seems that MC produce more ordered chains of PANI, which decorate the low conductivity TiO_2_ NPs. Since TiO_2_ is photoactive, the nanocomposite is used to build hybrid (TiO_2_/PANI NF) solar cells. Rajappa et al., uses MCP to produce a NC including Fe_2_O_3_ nanoparticles (FeOxNP) and a photoactive dye (tetraaminophthalocyanine (Zn),TAPcZn) [134]. The PANI covered the FeOxNPs and holds the photoactive dye. The NC shows excellent activity for the photodegradation of MB in water. Valadbeigi et al. uses MCP (with FeCl_3_·6H_2_O as an oxidant) to produce a nanocomposite of poly(o-toluidine) and WO_3_ nanoparticles [135]. As in other MCP with FeCl_3_.6H_2_O [28], they observe that the heat produced by the reaction melts the hydrated iron chloride (m.p. = 37 °C), and most of the polymerization occurs in solution (with fused salt as a solvent). The NC shows a core of WO_3_ NPs surrounded by a shell of POT. The NC shows photovoltaic activity, and solar cells were built. 

-Nanocomposites with metallic particles

Farrage et al. use MCP to produce PANI by oxidation of aniline with Ag+ ions, which became reduced to Ag nanoparticles [87]. TEM images show PANI nanoparticles (200–250 nm dia) with much smaller (<10 nm) Ag NPs uniformly distributed in the PANI NP. Paulraj et al. polymerize oPD using Ag^+^ (AgNO_3_) as an oxidant (E° = Ag^+^/Ag = 0.7996 V_NHE_) [138]. oPD oxidizes (E^o^ = 0.46 V_NHE_ [143]) and polymerizes, while Ag^+^ is reduced to Ag^0^, forming nanoparticles. The FESEM shows poly(oPD) particles of ca. 400 nm dia with small (<15 nm) Ag NPs evenly distributed in the polymer mass. The NCs were used as electrochemical sensors for the oxidation of hydrazine and reduction of hydrogen peroxide. 

-Nanocomposites with other substances

Ding et al., produce a nanocomposite of β-CD and PANI [119]. Depending on the proportions between components, the shape of nanostructures changed. For 80/20 (aniline/β-CD) rod-like structures are observed, Moreover, the conductivity of the 80/20 nanocomposite has higher conductivity (6.3 S/cm) than PANI without β-CD (5.1 S/cm).

## 3. Conclusions

Mechanochemical synthesis of polyaniline and substituted polyanilines is an ecofriendly tool to produce conducting polymers without producing large amounts of contaminated solvent (water) and/or using large reactors to hold diluted (<0.2 M) solutions. While PANIs can be formed using FeCl_3_ as an oxidant, the most widely used oxidant is ammonium peroxydisulfate (APS). The stoichiometry of the polymerization reaction requires 1.25 moles of APS per aniline ring in PANI (2.5 moles of e^−^). In mechanochemical polymerization (MCP) of anilines, it seems that a lower oxidant/aniline ratio produces material with higher conductivity and molecular weight. The polymerizations of substituted anilines, which produce linear polymers, are quite similar to that of aniline. On the other hand, anilines substituted with –NH_2_ or –OH groups (in *ortho* and *meta* positions on the amino group in aniline) lead to ladder polymers containing phenazine or phenoxazine rings. In that case, the amount of oxidant should be larger, since the intramolecular attack requires more electrons. All the polymerizations produce protons (two per attack) in the formation of bonds with the aromatic rings. In all cases, salt oxidants (APS or FeCl_3_) produce anions that form acid with the protons released during the electrophilic substitution. In a polymerization solution, the acids dissolve in the solvent (albeit lowering the pH), but in a solid state reaction remain in the mixture and could be used to dope the conductive form of the polymer. Moreover, volatile acids (e.g., HCl) could be released from the mixture. Experimental data are presented, suggesting that HCl is indeed released when aniline hydrochloride is used as a monomer. The known hazard of HCl gas [144] should be taken into account in the operation of the technique in open vessels (e.g., mortar). In the case of closed reactors (e.g., ball mill jars), consideration for buildup pressure and management of the gas has to be taken into account. The amount of water present during the grinding/milling of reaction mixtures is unknown, taking into account that APS or anilinium salts are very hygroscopic. Some of the published experimental procedures leave the reactant mixture for days in air to complete the reaction. It is likely that water absorption plays a role in the completion of the reaction. It should be advisable to maintain a high and constant humidity during such resting period to improve the reproducibility. Others wash the product of MCP with water (or aqueous solutions). Finally, some researchers add water directly, or use a solution such as 37% HCl, to the reactant mixture. This addition increases the polymerization rate but, in fact, converts the solid state reaction into a solution reaction at high concentration. The drawback is that the oxidative polymerization of aniline is highly exothermic. A simulation using the kinetic model predicts fast reactions (<60 s) and high temperatures (>300 °C). A quick experiment, using a nearly saturated solution of APS and AniHCl, shows a fast reaction (<2 min) with a large liberation of heat. The temperature increases until water boiling point is reached, showing the danger of working with concentrated solutions. Most of the work has been performed in open mortar and pestle devices. The grinding is usually made until a noticeable color change is observed, but then a rest period (in air without humidity control) of 1–14 days allows the reaction to continue, even with the action of oxygen from air. On the other hand, it is found that toxic fumes (aniline) are released, and protective gear should be worn. The MCP method could be applied to substituted anilines, producing material similar to PANI. A special case is the homopolymerization of 3-aminobenzensulfonic acid, which produces a polymer by MCP while only producing oligomers in solution. The polymerization is made using a fused salt (FeCl_3_·6H_2_O) as both oxidant and solvent. MCP below the melting point of the salt produces low yield, while the fused salt gives a good yield (46%). To the best of our knowledge, this is the only case of mechano-exclusive polymerization of a CP (i.e., a synthesis that can only be performed by MCP). 

Polymerization of o-aminophenol and o-phenylenediamine seems to produce ladder polymers by a cyclization step of the linear polymer. Diphenylamine has been polymerized by MC by tail-to-tail dimerization and chain formation. 

A large body of work has been devoted to the preparation of nanocomposites by MCP of aniline in the presence of other materials. Several works deal with different kinds of clays where PANI grow inside the galleries of the natural material. The clay protects intercalated PANI, improving the thermal stability. The conductivity of the nanocomposite is lower than PANI but large enough for different applications. Nanocomposites of carbon materials (nanofibers, single and multiwall nanotubes, and graphene) have been made by MCP of anilines where the milling also disperse the phases and/or exfoliate the carbon material. The carbon gives surface area and conductivity, and the nanocomposites show large capacities of electrochemical charge storage. The materials are promising for supercapacitor applications. Nanocomposites made of metal nanoparticles dispersed in polyanilines were made by MCP of silver salts mixed with anilines. The monomer is oxidized by Ag^+^, and Ag nucleates and grows to form nanoparticles. In that way, solid dispersions of small Ag NPs in larger polymer nanoparticles are produced. MCP of aniline in the presence of oxides (Fe_2_O_3_, TiO_2_, or WO_3_) and other photoactive species allow one to build photovoltaic electrodes. 

## 4. Future Outlook

The parameters of aniline MCP, such as the role of water/liquids and temperature, should be studied in detail. Using mechanized milling devices (e.g., ball mill), it would be possible to study the effect of milling parameters (material of ball and jars, number and size of the balls, frequency of rotation, etc.) on the MCP of anilines, as already suggested by Krisenbaum et al. [23]. Such a study has been performed for the Gilch synthesis of poly(phenylene vinylene) [145]. Moreover, using a closed MC reactor, the atmosphere of the reaction (humidity and oxygen content) could also be controlled, and exposure to toxic volatiles could be avoided. There are other conditions where MCP could be at an advantage compared with polymerization in solution. Beadle et al. polymerize aniline at low temperatures (down to −40 °C) and show that PANI with high conductivity (up to two orders of magnitude) can be produced [146]. It seems that the oxidation reaction of aniline (or the growing chain) by persulfate is little affected by temperature. However, to maintain the aqueous/alcoholic solution liquid, they have to add up to 2 M LiCl, and it has been shown that the presence of a high Cl^−^ concentration induces chlorine incorporation in the chain [23]. On the other hand, MCP of dry aniline salts could be made at any temperature without additives. Moreover, Konyushenko et al. show that aniline can be polymerized with frozen solutions [46]. Therefore, MCP of frozen aniline at low enough temperatures (<−40 °C) could produce PANI with improved properties. In 2019 IUPAC identified reactive extrusion, a form of MC, as one of the 10 chemical innovations to change our world [147]. The technique uses a solid powder (or paste) flow reactor, which can produce industrially-relevant quantities by continuous production. The short polymerization time of PANI by MCP (<5 min) should allow short residence times and fast production. Moreover, since the dry mixture of anilinium salt and oxidant are stable before grinding, a single screw extruder (common in polymer processing) can be used. While great pains have been taken to use solid reactants, pure aniline could be used with solid oxidants, forming a paste. A small amount of dry solid acid (e.g., toluenesulfonic acid) will maintain the reaction acid until the protons produced in the reaction give an acid medium. 

### 4.1. Synthesis of PANI Doped with Polyelectrolytes

Polyelectrolytes, which are soluble in water (e.g., PSS), have been used to disperse/solubilize PANI by polymerizing aniline in the presence of the anionic form at a neutral pH. To do that, the polymerization has to be carried out by enzymatic polymerization [148]. Using MCP, it is possible to make PANI in the presence of the acid form (e.g., PSSH) either by previous formation of the PSS-ANI salt in solution or using AniHCl. Then, the PSS-PANI is dissolved in basic/neutral pH where the parts of the PSS chain that do not interact with PANI dissolve in water (ion-dipole interactions) and solubilize/disperse the PSS-PANI. The Donnan effect of –SO_3_^−^ precludes the protons to leave [149]. Therefore, the PSS-PANI remains in its doped (conductive) state. While sulfonic acid groups could clearly protonate PANI, it has been shown that carboxylic groups, aided by Donnan effects, are also able to maintain PANI in the protonated state [150]. Therefore, biobased polymers (CMC, CMS, CHI, or CHSu) or biomolecules (HA, proteins) can also form complexes or nanocomposites with PANI. 

### 4.2. Doping with Acids/Anions Insoluble in Water

Since polyaniline is not soluble but is wetted by water it is easily doped/dedoped by small ions/acids immersion the solid on aqueous solutions. On the other hand, doping with acids insoluble in water (e.g., stearic) is difficult. Moreover, the incorporation of a hydrophobic counterion is required to produce more hydrophobic doped polyaniline. Mechanical grinding of the solid acid with PANI (EB) could be used to produce doped PANI (ES) with hydrophobic counterions. 

### 4.3. Synthesis of Oxidized/Reduced Forms

PANI is stable in air in the half-oxidized (emeraldine) form. To produce the other forms (lecucoemeraldine and pernigraniline), the polymer has to be reacted with a reductant/oxidant. Usually, such reactions are performed by immersing the solid in solutions. However, the reaction occurs heterogeneously. Using solid reductants (e.g., ascorbic acid) would possibly reduce PANI in a dry state, where reoxidation by air is slower than in solution. In fact, a mechanochemical procedure for the reduction of PANI was described by Epstein and coworkers to produce highly sulfonated PAN [151], and used by Barbero et al. to introduce amide groups in the nitrogen of PANI (LEB) [152]. Mechanochemical oxidation to the fully oxidized form (PNB or PN) could be produced using solid dry persulfate salts. The absence of water avoids degradation by nucleophilic attack. 

### 4.4. Synthesis of Functionalized Polyanilines

Polyanilines with functional groups attached to the main chain have been synthesized in solution and show better processability (solubility) and new properties, which lead to different technological applications [153]. The materials can be produced by homo/copolymerization of substituted monomers or PANI post-functionalization.

-Homopolymerization. Anilines monosubstituted in nitrogen (e.g., N-methylaniline) or bearing small electron donating groups in the ring (e.g., 2-ethylaniline) can be homopolymerized. On the other hand, anilines with electron withdrawing groups (e.g., –SO_3_^−^) only produce oligomers. Remarkably, 2-aminobenzensulfonic acid was polymerized in a fused salt (FeCl_3_.6H_2_O), which acts as solvent and oxidant [28]. The method could be applied to other aniline monomers bearing electron withdrawing groups (e.g., 2-nitroaniline).-Copolymerization. Copolymerization of an unreactive aniline (e.g., 2-aminobenzoic acid) with aniline renders copolymers in solution [154]. However, the lower reactivity of the substituted aniline compared to aniline makes the polymerization slower and creates compositional shift. The data on MCP of aniline suggest that stoichiometric control is more important than kinetics. Therefore, it could be possible to produce copolymers with more a defined composition ratio using MCP of the monomer mixture.-Post-functionalization. Direct reaction of PANI chains allows for attaching functional groups to the polymer. The most used reaction (sulfonation) is an electrophilic aromatic substitution reaction and is usually made in homogeneous solution by reacting PANI with SO_3_ with the polymer dissolved in concentrated H_2_SO_4_. However, Epstein and coworkers used a thermal reaction of PANI with SO_4_ (NH_4_)_2_ to sulfonate PANI [155]. The same reactant, or stronger solid sulfonating agents (e.g., SO_3_.Pyridine complex), could be used for MC functionalization of PANI. On the other hand, nucleophilic addition to PANI is usually performed heterogeneously [156]. Sulfonated polyaniline can be produced by nucleophilic addition of sulfite ions [157]. Using solid reactants, it is possible to achieve MC functionalization of PANI.

### 4.5. Fabrication of PANI Nano-Objects

There is evidence that PANI produced by MCP is produced as solid aggregation of nanospheres (Figure 4) or nanofibers [74,75,76]. Using MC and/or other deaggregation methods (e.g., ultrasound), together with stabilizers of colloids (DBSA, PVP, PSS), will be a possible way to produce stable dispersions of PANIs nanoparticles. 

### 4.6. Synthesis of PANI Containing Nanocomposites

As shown above, the synthesis of NC by MCP of anilines in the presence of solid materials (e.g., clay) is a flourishing field of study. However, several venues for future work exist. MCP polymerization with graphene oxide (GO) or reduced graphene oxide (rGO) could produce materials comparable with the NC made from graphite, albeit with covalent linking of the reactive oxides with PANI. 2D layered materials differentfrom graphene (e.g., MoS_2_ layers) can also be used to produce NCs by MCP of anilines [158]. Large surfaces such as porous carbon materials can be either produced from lignocellulosic biomass [159], by pyrolysis and activation, or by the pyrolysis of synthetic gels [160]. Porous materials such as metal organic frameworks (MOF), covalent organic frameworks (COF), polymer organic frameworks (POF), porous disordered (e.g., silica gel), and mesoporous ordered (e.g., SBA 15) oxides can be made into nanocomposites with PANis using in situ MCP. The polymer chains could grow inside the cavities/channels in the porous structures. In that way, the dispersion will be complete and the matrix could protect the polymer against degradation. Moreover, growing PANI inside athe interpenetrated porous structure will generate a porous PANI solid intertwined with the porous matrix. If the matrix could be dissolved (e.g., HF) or disassembled (e.g., changing pH), the porous PANi solid could remain in place. Such a high surface area conductive framework could have plenty of applications including supercapacitors/batteries, water remediation, catalysis, etc.

### 4.7. Using MCP for 2D Printing of PANI Thick Films on Fabrics

As discussed above, an early example of MC is the formation of “aniline blacks” [64]. While so called “ungreenable aniline blacks” likely contain phenazine units (linear emeraldine is green) and are poor conductors, the synthesis form the linear PANI, which is then converted to aniline black [31]. An analysis of Lightfoot’s recipe [161] shows that AniHCl oxidizes with chlorate as an oxidant and copper sulfate as the catalyst. Additionally, using 95 % in weight of starch produces a thick paste. The amount of copper salts is ca. 20% of the chlorate suggesting a contribution as an oxidant. Assuming reduction of chlorate (ClO_3_^−^) to chloride (Cl^−^) and of Cu^+2^ to Cu^0^, the amount of oxidants are twice that necessary for PANI (ES) formation. Therefore, the polymer is likely to be overoxidized to form aniline black [162]. Using a stoichiometry amount of oxidant (likely APS) would be possible for precipitating PANI (ES) on cellulosic fibers (e.g., cotton). While other procedures are known to produce PANI films on solids [82], they made thin (<300 nm) films, while thick deposits (>10 μm) can be made by this method. The PANI could be patterned by pressuring a stamp on certain regions of the pasted fabric, effectively 2D printing by MCP. Other options, like “drawing” with a rolling ball pressing the fabric at defined positions would allow for customized patterning. Those thick conductive patterns can be used directly for flexible electronics or used to deposit metals (e.g., Cu) to produce printed circuits. Moreover, thick macroporous films, made of fibers covered with thick films of PANI, could be used in applications like supercapacitors, batteries, or disposable electrodes. In those cases, the fabric should be also conductive, such as a fabric made of carbon fibers.

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
