# Peer review of "Mechanochemical Synthesis of Polyanilines and Their Nanocomposites: A Critical Review"

_polymers, 2022, doi:10.3390/polym15010133_

Round 1

Reviewer 1 Report

The present manuscript deals with an interesting aspect of polymer chemistry, namely the mechanochemical polymerization. Despite this interesting topic for many potentials readers, I fear that the manuscript is unsuited for publication in its present state for several reasons:

First of all, a clear definition of "Mechanochemical processes" is not given at all! How is a mechanochemical reaction defined? A hint in the Introduction section is clearly missing!

Abstract: There are contradicting statements in the abstract, i.e. First line the authors state: “The mechanochemical synthesis of polyanilines, by oxidative polymerization of anilines … is reviewed.” In the last line of abstract: “The extensive work on the production of nanocomposites by MCP of anilines together with different nanomaterials … is described.” So what is true?

Besides, there are incomplete sentences, missing brackets, etc in the abstract.

Introduction section: The authors quote 0.1-1 M as a low concentration, which is in my mind a quite concentrated solution …Please specify!

Then: Synthesis of polymer uses a lot of solvent which becomes contaminated (e.g. water), or it is a contaminant itself? The polymer is a contamination? Unclear …

Please explain acronyms when used first (MC), page 2, line 50. 4ADA, page 2, line 70., PANI – page 2, line 81, DMSO, page 3, line 114, CAN, page 3, line 118, ES-form, page 3, line 120, APS, page 4, line 163, etc. A list of symbols at the end of the manuscript is not very helpful for the reader ...

Many incomplete sentences such as: “One solution is the use mechanochemical synthesis where two solids are mixed …” Page 2, line 46.

“The oxidative formation of dyes an precipitates by oxidation of aniline even before Staudinger’s macromolecular hypothesis.” Page 2, line 81-83., etc.

Results – Section 2.1 – please provide English title!

Page 3, lines 98-103: For me, electrochemical oxidative polymerization is the most obvious “oxidative” polymerization. No literature is provided here. Please complete.

Weird sentence constructions and incomplete sentences that do not make sense and are hard to understand: e.g. page 3, line 108 “The aniline catión radical suffer follow up irreversible reactions in ms …” or page 3, line 128/129: “Since the cation radical is less acidic than anilinium, will be only deprotonated in neutral media.” Page 5, lines 144/145 “On the other hand, the side chain act as crosslinks increasing the molecular weight.” Page 6, lines 169/170 “Since mechanochemical growth of PANI in the solid state of PANI, could have more in common with electrochemical polymerization than with chemical polymerization. “ Many text passages are unclear, and hard to understand, even after several readings.

Unclear, why does benzidine concentration increases at lower pH. (page 3, line 130) Rather speculative!

Schemes, and Figures are partly not explicitly mentioned in the text, so their presentation appears unnecessary, although they are required for the understanding of the manuscript. E.g. what is the use of Scheme 2, and Scheme 3? Please improve.

Page 6, line 200, the link should be to Figure 1, not to Figure 2.

Figure 1, the “theory” is entirely unexplained. How it is justified to extrapolate to a temperature of 580°C or 600°C? How precise are these estimates? How is the theory justified? Much more explanations and discussion are needed here!

Mixed singular/plural cases e.g. page 8, 236, “… the chains remains in the emeraldine state.”

Page 8, Line 244, VNHE should read VNHE, i.e. in relation to the standard hydrogen electrode, I guess – or it is a new abbreviation?

Page 10, line 289, and 290, it should read “silicotungstic acid“ instead of “silicotunsgtic acid”

To make Figure 2 more clear, please use Figure 2(a) and Figure 2(b) and relate the text accordingly. Why are the observed behaviours so different? The observations are not explained or motivated by the authors.

Page 14, first two lines: I feel it is a bit dangerous to conclude that a reaction scheme is true simply by using a simple pH test above the mortar – this appears rather speculative to me! Much more quantitative tools / techniques and additional information (FTIR, etc) need to be included to do so. Please comment.

Page 14, first paragraph: What do the authors understand by “doping”? This is a well-defined phenomenon in physics and chemistry, and also in organic/polymer chemistry, and should be used accordingly. Please be more specific!

Organisation of the manuscript in section 2: Why do the authors introduce section 2.1.1. etc? 2.1.4.1? The structure needs to be substantially improved.

I found many of the explanations given throughout the manuscript, very, very speculative, and not supported by the data shown.Although the authors collected a wealth of information (i.e. more than 140 references), they did not discuss these works adequately, and so the claim to be a Review article is not at all fulfilled. A much more structured and detailed discussion is required in my opinion. Therefore, I would propose to reject the manuscript, it does not justify its publication as a review in "Polymers"!

Author Response

The present manuscript deals with an interesting aspect of polymer chemistry, namely the mechanochemical polymerization. Despite this interesting topic for many potentials readers, I fear that the manuscript is unsuited for publication in its present state for several reasons:

 First of all, a clear definition of "Mechanochemical processes" is not given at all! How is a mechanochemical reaction defined? A hint in the Introduction section is clearly missing!

 A whole section is added to clarify the point

Abstract: There are contradicting statements in the abstract, i.e. First line the authors state: “The mechanochemical synthesis of polyanilines, by oxidative polymerization of anilines … is reviewed.” In the last line of abstract: “The extensive work on the production of nanocomposites by MCP of anilines together with different nanomaterials … is described.” So what is true?

Both, the first phrase deal with polyanilines and the last one with nanocomposites of polyanilines with different nanomaterials. The last phrase is rewritten to further clarify it.

Besides, there are incomplete sentences, missing brackets, etc in the abstract.

The abstract is rewritten and the typos corrected.

Introduction section: The authors quote 0.1-1 M as a low concentration, which is in my mind a quite concentrated solution …Please specify!

Corrected

Then: Synthesis of polymer uses a lot of solvent which becomes contaminated (e.g. water), or it is a contaminant itself? The polymer is a contamination? Unclear …

The byproducts (e.g. benzidine which is a known cancerigen) are contaminants

Please explain acronyms when used first (MC), page 2, line 50. 4ADA, page 2, line 70., PANI – page 2, line 81, DMSO, page 3, line 114, CAN, page 3, line 118, ES-form, page 3, line 120, APS, page 4, line 163, etc. A list of symbols at the end of the manuscript is not very helpful for the reader ...

The abbreviation list is moved to the begining

Many incomplete sentences such as: “One solution is the use mechanochemical synthesis where two solids are mixed …” Page 2, line 46.

Corrected

“The oxidative formation of dyes an precipitates by oxidation of aniline even before Staudinger’s macromolecular hypothesis.” Page 2, line 81-83., etc.

Corrected

Results – Section 2.1 – please provide English title!

We do not understand the comment

Page 3, lines 98-103: For me, electrochemical oxidative polymerization is the most obvious “oxidative” polymerization. No literature is provided here. Please complete.

A reference is included

Weird sentence constructions and incomplete sentences that do not make sense and are hard to understand: e.g. page 3, line 108 “The aniline catión radical suffer follow up irreversible reactions in ms …”

Corrected

or page 3, line 128/129: “Since the cation radical is less acidic than anilinium, will be only deprotonated in neutral media.”

Rephrased

Page 5, lines 144/145 “On the other hand, the side chain act as crosslinks increasing the molecular weight.”

Rephrased

Page 6, lines 169/170 “Since mechanochemical growth of PANI in the solid state of PANI, could have more in common with electrochemical polymerization than with chemical polymerization. “

Deleted

Many text passages are unclear, and hard to understand, even after several readings.

Unclear, why does benzidine concentration increases at lower pH. (page 3, line 130) Rather speculative!

Reference added

Schemes, and Figures are partly not explicitly mentioned in the text, so their presentation appears unnecessary, although they are required for the understanding of the manuscript. E.g. what is the use of Scheme 2, and Scheme 3? Please improve.

The schemes show the reactions which occur during the different steps of polymerization in solution, which are likely to occur during MCP but were not discussed in the reviewed work. Therefore, we consider them useful. The schemes and figures are mentioned in the text, some (Scheme 2) more than others.

Page 6, line 200, the link should be to Figure 1, not to Figure 2.

Corrected

Figure 1, the “theory” is entirely unexplained. How it is justified to extrapolate to a temperature of 580°C or 600°C? How precise are these estimates? How is the theory justified? Much more explanations and discussion are needed here!

The simulation was discussed in our previous publication [50], where the whole curve (under cooling and low concentration of reactants) was simulated. Obviously other reactions will occur above 100 oC  but the point that high local heat could develop is clearly shown by the experiment. Now, another experiment where some temperature increase is observed during mortar grinding of ANI.HCl and APS. A phrase is added to contextualize the results

Mixed singular/plural cases e.g. page 8, 236, “… the chains remains in the emeraldine state.”

Corrected

Page 8, Line 244, VNHE should read VNHE, i.e. in relation to the standard hydrogen electrode, I guess – or it is a new abbreviation?

Corrected

Page 10, line 289, and 290, it should read “silicotungstic acid“ instead of “silicotunsgtic acid”

Corrected

To make Figure 2 more clear, please use Figure 2(a) and Figure 2(b) and relate the text accordingly. Why are the observed behaviours so different? The observations are not explained or motivated by the authors.

The figures are separated and new Figure 4 placed close to the discussion of the difference

Page 14, first two lines: I feel it is a bit dangerous to conclude that a reaction scheme is true simply by using a simple pH test above the mortar – this appears rather speculative to me! Much more quantitative tools / techniques and additional information (FTIR, etc) need to be included to do so. Please comment.

The mechanism seems true because there is a large amount of work in solution. Moreover, pH decrease during polymerization has been recorded extensively. However, in MCP most authors (not all, see [62]) are oblivious to the fact that acid is produced during polymerization. In the case of MCP of ANI,HCl has not been recorded before and could be quite dangerous. The phrase is corrected.

Page 14, first paragraph: What do the authors understand by “doping”? This is a well-defined phenomenon in physics and chemistry, and also in organic/polymer chemistry, and should be used accordingly. Please be more specific!

Doping is defined in conducting polymers as the amount of counterions present to balance the mobile charges, created by oxidation. In polyaniline acid doping is defined which involves the creation of charges by protonation of the quinonimine. This is the accepted meaning and the one used trough the manuscript. Doping% is used in table 4 as the % of counterions (elemental analysis) or ratio of IR bands.

Organisation of the manuscript in section 2: Why do the authors introduce section 2.1.1. etc? 2.1.4.1? The structure needs to be substantially improved.

The structure is suggested in the Template of Polymers and divide the work by similar monomer or nanomaterial (for composites)

I found many of the explanations given throughout the manuscript, very, very speculative, and not supported by the data shown. Although the authors collected a wealth of information (i.e. more than 140 references), they did not discuss these works adequately, and so the claim to be a Review article is not at all fulfilled.

On the contrary, the reading/analysis has been very exhaustive. In the process, we have recalculated results from the published figures (Figure 3 and 4), extract detailed experimental details from the whole text of the manuscripts (Tables 1, 2, 3) and even (when possible) calculate the molar ratio when only weights/volumes are provided. Moreover, we simulate and measure the temperature in an experimental condition already reported (addition of water) and analyzes the thermal data of polymerization at high concentrations (Figure 1) to put the MCP data in context. Now, we measure the temperature during mortar grinding. The breadth of the detailed analysis of the data is now more clearly described in the text and captions.

It should be mention that not all 140 references describe MCP work but several describe chemical and electrochemical polymerization in solution along with proposals of reactions, mechanism and/or kinetic data. The purpose of those references is to give the context to critically analyze the MCP data, specially the experimental conditions. Such references are well established work, most of them with more than 100 citations, and are not analyzed but used as input for the analysis of MCP data. Since the lead author of the review has more than 30 years’ experience in the field (and more than 70 published works in it) the models described in those references are part of the accepted knowledge we used to write the review. It seems that the reviewer considers stating an accepted reaction or mechanism supported by previous work a speculation.

A much more structured and detailed discussion is required in my opinion. Therefore, I would propose to reject the manuscript, it does not justify its publication as a review in "Polymers"!

The reviewer is entitled to his/her opinion but see as an example, the analysis of a single reference:

“Abdiryim et al, studied the effect of different strong organic acids (TSA, MSA, DBSA) on the properties of solid-state synthesis of PANI [67]. The procedure is different than Kaner and coworkers [66]. One ml of water is placed on a mortar, then pure acid (0.015 mol,15 M) is dissolved and aniline (0.01 mol, 10 M) is added. As it can be seen, there is a ca. 50 % excess of acid. A white precipitate is formed, which is the anilinium salt of the acid since the concentration is larger than the solubility of the salt. Then, persulfate (2.2 g, 0.0097 moles) is added and grinding with the mortar during only 30 min carried out the reaction. The ratio aniline/oxidant is close to 1.1. As it was discussed before, the reaction does not likely occur in dry state but in concentrated (unknown) solution of anilinium salt and persulfate (maximum solubility 3.5 M). Therefore, it will be very fast (tR = 60-100 s) and with a large thermal heating. Indeed, the cyclic voltammograms of PANI produced using this method show a significant “middle peak” which is usually assigned to redox groups (e.g. quinones [61]), produced during PANI degradation.”

We feel that this is a quite detailed discussion. About the structure, we divide the manuscript in different kinds of monomers: aniline, substituted anilines, special o-substituted anilines and diphenylamines. The order is related to the mechanism of polymerization. Then, the extensive work on MCP of nanocomposites is described. In this case, the kind of nanomaterial used defines the different sections. The published work is quite heterogeneous and we feel this structure allow discussing the different parameters affecting MCP of anilines. We feel this is much better than a long enumeration of different work with no order.

Reviewer 2 Report

This review provides a comprehensive overview of advances achieved in the field of

mechanochemical synthesis of polyanilines. 

The outlook for the future should be supplemented. 

There are some grammatical mistakes in the article, which should be corrected.

Author Response

Reviewer 2

his review provides a comprehensive overview of advances achieved in the field of

mechanochemical synthesis of polyanilines. 

The outlook for the future should be supplemented. 

A whole Future Outlook section is added

There are some grammatical mistakes in the article, which should be corrected.

The manuscript has been thoroughly checked and the grammatical errors corrected.

Reviewer 3 Report

More introduction about the mechanochemical polymerization should be added.

The chemical equation shown in Scheme 4 is nonstandard.

Supercapacitor is an important application direction for polyaniline. This part should be summarized.

The SEM image to indicated the microstructure of polyaniline and its derivants should be included in the text.

Author Response

Reviewer 3

More introduction about the mechanochemical polymerization should be added.

A whole new long paragraph is added

The chemical equation shown in Scheme 4 is nonstandard.

Since several acids an oxidants are used in the work reviewed, we have to show a general reaction which describes all different conditions

Supercapacitor is an important application direction for polyaniline. This part should be summarized.

We agree, the discussion of those works where materials useful in supercapacitors were synthesized were extended.

The SEM image to indicated the microstructure of polyaniline and its derivants should be included in the text.

We include a Figure where the microstructure of PANI produced in solution and by MCP is compared. The data is analyzed in full.

Round 2

Reviewer 3 Report

All the issues have been addressed.

Author Response

Thank you for the careful reading. All the mistakes are corrected.

Line 239 Synthesis de polyanilines ( Synthesis of polyanilines)

Corrected
Line 272: “Catión” (cation)

Corrected
Line 278: dimerizarion (dimerization ?)

Corrected
Line 285: trough (through ?)

Corrected
Line 321: suggest (suggests) (“The long discussion ….suggests”

Corrected